# OpenReview forum: "Mass-Producing Failures of Multimodal Systems with Language Models"
_NeurIPS.cc/2023/Conference — NeurIPS 2023 poster_

### Official Review · Reviewer_ATN5 · 2023-07-01

**Soundness:** 3 good
**Presentation:** 4 excellent
**Contribution:** 3 good
**Rating:** 7
**Confidence:** 4

**Summary:**

The authors present MultiMon, which is a system for automatically identifying both general example categories that multimodal models struggle with, and new specific examples that would likely produce errors. MultiMon does this by using a semantic-similarity text encoder to find sufficiently different sentences in a corpus, and then giving these sentences to CLIP to see if CLIP actually produces very similar embeddings for them. If CLIP does produce similar embeddings for sufficiently different sentences, then this is evidence that a variety of multimodal models may not be able to distinguish them. MultiMon gives these sentences to powerful LLMs such as GPT-4, and asks the LLMs to identify general categories of examples that multimodal models may struggle with. The LLMs are then asked to generate specific examples of each category that they have identified. Overall the output of MultiMon is challenging test datasets with novel examples for multimodal systems. Subsets of these challenging test datasets have been verified by the authors to produce a significant number of failures on downstream text-to-image, text-to-video, and text-to-3d-model systems.

**Strengths:**

The writing and presentation is clear and fairly comprehensive, and the claims seem well supported by the evidence. As far as I am aware, the work is sufficiently original. They present an important general idea, which is to bootstrap multimodal failure knowledge from incredibly proficient text-only LLMs in an automatic way. I think that approaches like this one will only become more important in the field as LLMs improve even more.

**Weaknesses:**

The authors used themselves as annotators to evaluate how well their failure-finding procedure could actually find real failures on downstream generation tasks. It could be useful to get independent annotators who are not invested in the project though. It shouldn't be terribly expensive to spin up a task on MTurk - the authors only checked 100 pairs.

I found myself really relying on the more meaty figures in the appendix to validate the claims in the paper. You could consider at least putting this in the main text: a table of downstream failure rates of various models as judged by human annotators (ideally not the authors). It will help people skimming the paper who just want to see how well your approach actually works.

I see that you can automatically identify failures, but can you automatically fix them? Did you try incorporating the generated data into training routines? How did it do? You could at least use this automatic-fixing end goal to help motivate the paper if you have room.

Hugging Face is two words, both capitalized.

**Questions:**

Did you test MultiMon on any vision models that definitely don't use CLIP under the hood? It would be great if your failure-finding procedure could be shown to generalize beyond CLIP-based models.

The 100 pairs that you manually checked are randomly sampled, right?

Are the discovered failure categories in Appendix C quoted verbatim from the language models? It is unclear to me. It would also be helpful to include a specific example generated by the model for each class it discovered in Appendix C.

**Limitations:**

I believe that the limitations are adequately addressed.

---

> ### Author Rebuttal · Authors · 2023-08-09
>
> We thank the reviewer for the comprehensive review and valuable feedback! We appreciate that you find our work “present[s] an important general idea” and think “our approach will only become more important”. We include explanations below to address your points, including new experiments on systems that do not use CLIP:
>
> ---
>
> *Did you test MultiMon on any vision models that definitely don't use CLIP under the hood? It would be great if your failure-finding procedure could be shown to generalize beyond CLIP-based models.*
>
> Thank you for your suggestion! Based on it, we test how well MultiMon can find failures on multimodal models that definitely do not use CLIP in two settings:
> 1. **Transfer**. We show that the inputs we find using CLIP often directly produce failures in T5-based text-to-image models; 70.8% of the inputs used to generate figures in the main body or the appendix produce failures on DeepFloyd, a text-to-image model that uses T5 to embed inputs, rather than CLIP. [Figure 1, supplementary material]
> 2. **Generation from scratch**. We apply MultiMon on the T5 model from scratch (i.e., replace CLIP with T5 throughout the whole pipeline) and find some systematic failures of the T5-based model that the models we study do not have [Figure 2, supplementary material].
>
> We go into further on the generation experiment below.
>
> **Generation details**. To find systematic failures of DeepFloyd, from scratch, we repeat the entire MultiMon pipeline from Section 3, using T5 embeddings instead of CLIP (using our existing code, this took under three hours.) We find many overlapping systematic failures but some new ones. For example, a new systematic failure that MultiMon outputs is:
>
> - _The model fails to comprehend the use of pronouns properly. The sentences are similar, but the change of subject affects the visual representation significantly_
> - _The model fails to distinguish the time of day. This is critical in visual representation, as these times would significantly change the lighting, color scheme, and potentially the activity depicted in the image_
>
> Overall, the systematic failures we find with MultiMon on T5 have an average success rate of 77.3%. We have also attached some inputs that MultiMon generates using T5 along with their images generated with DeepFloyd [Figure 2, supplementary material].
>
> ---
>
> *The authors used themselves as annotators to evaluate how well their failure-finding procedure could actually find real failures on downstream generation tasks. It could be useful to get independent annotators who are not invested in the project though. It shouldn't be terribly expensive to spin up a task on MTurk - the authors only checked 100 pairs.*
>
>
> Thanks for raising this point! We definitely think there are risks in having authors invested in the project doing labeling, so we designed our study to be deliberately *not* gameable. Each author labeled 400 chosen images: 100 random pairs of images from MultiMon, and 100 random pairs of images from the baseline system. However, we scrambled these 200 pairs together, so the authors were not given information on whether each pair was from the baseline versus from MultiMon (and thus couldn’t game which input to select, or exploit the “no match” option). Since there was a significant gap in accuracy between the baselines (80%) and MultiMon (20%), we expect that the failures are genuine. We will clarify this setup in subsequent versions of Section 5.1.
>
> We also agree that setting up a MTurk task would add further validation. We were not able to get IRB approval for the study in time for the rebuttal, but expect the results to be very similar to our study and will consider adding it to subsequent versions of our work.
>
> ---
>
> *It will help the readers to put more meaty figures from appendix to the paper to validate the claims in the paper.*
>
> We will update based on your suggestion; thanks so much for reading our paper so carefully!
>
> ---
>
> *Can you automatically fix the failures you identified? You should at least use this automatic fixing end goal to help motivate the paper.*
>
> Ultimately, we hope that developers can use MultiMon to improve subsequent generative systems, e.g., by using it as a source of examples for adversarial training. However, fixing these examples requires not only retraining CLIP, but retraining the entire diffusion model to adjust to the new embeddings [300 - 301]. We will discuss this motivation in the discussion of subsequent versions of our work.
>
> ---
>
> *Hugging Face is two words, both capitalized*
>
> Thank you for catching this mistake! We will change all the Hugging Face in our text.
>
> ---
>
> *The 100 pairs that you manually checked are randomly sampled, right?*
>
> Yes, randomly sampled. We’ll make sure to clarify this
>
> ---
>
> *Are the systematic failures in Appendix C verbatim from LLM? + It would be nice to include a specific example generated by the model for each class it discovered in Appendix C.*
>
> Yes, the descriptions for systematic failures are copied verbatim from LLM. And, thank you for the advice to put examples beside each class discovered.  We will update it in the Appendix in the revised version.

---

> > ### Comment · Reviewer_ATN5 · 2023-08-15
> > **Response**
> >
> > I really appreciate that you took the time to carefully respond to the critiques / questions and even ran some new experiments. I can't think of any more concerns at the moment. Definitely keeping my rating as a full "accept".

---

### Official Review · Reviewer_dXH6 · 2023-07-04

**Soundness:** 3 good
**Presentation:** 3 good
**Contribution:** 3 good
**Rating:** 6
**Confidence:** 4

**Summary:**

They propose MultiMon, a multimodal monitor, to automatically find, categorize, and generate failures of multimodal models. They categorize systematic failures of multimodal models using large language models and show that failures of CLIP embeddings also lead to failures in models that use these CLIP embeddings.

**Strengths:**

The authors propose an automatic evaluation pipeline for multimodal models.

- Given an initial corpus for evaluation, they leverage large language models (LLMs) to systematically categorize systematic failures in that corpus as well as use LLMs to generate more examples from each category.
- Their method can be steered for specific downstream applications like self-driving
- Their method has 3 stages each of which are plug and play and therefore, their proposed system is flexible
- They detect failures from an initial corpus without the need of generating outputs (though this might not be possible for models that don't have separate components that serve as a bottleneck for failures like CLIP embeddings do for current multimodal models)
- Their proposed system is able to generate examples that fool safety filters of MidJourney

**Weaknesses:**

- The proposed system heavily relies on the quality and availability of a curated initial corpus
- The authors do not provide any study on how the quality of the initial corpus affects the performance of their system
- Their system is easy to use for an adversary as well (dual use)
- The authors also do not provide examples of datasets that can be used as initial corpuses for their system (except for COCO that they use for experimentation)

**Questions:**

- As a suggestion, it would be good to quantify what is the minimum number of samples (and number of erroneous agreements expected, if any) that the initial corpus should have for their proposed system to actually provide a robust evaluation.
- Examples of existing datasets that can serve as good starters (apart from COCO that has been used in the paper) would also be very valuable for researchers looking to use this system

**Limitations:**

Yes they mention "We think deploying MULTIMON favors the evaluator over the adversary, as the evaluator gets to test for and fix the generated failures before release (at which point MULTIMON is useless to the adversary).". However, given the heavy reliance on the initial corpus on the types of failures, the proposed system generates, it's hard to be convinced that MultiMon is useless to the adversary (they can always use a better/diverse initial corpus) even if the evaluator had used the system to initially to do robust testing.

---

> ### Author Rebuttal · Authors · 2023-08-09
>
> Thank you for your valuable feedback and suggestions! We appreciate that you find MultiMon flexible, steerable and effective. We address your questions and comments below.
>
> ---
>
> *As a suggestion, it would be good to quantify what is the minimum number of samples (and number of erroneous agreements expected, if any) that the initial corpus should have for their proposed system to actually provide a robust evaluation.*
>
> Thanks for your suggestion; based on it, we quantify the properties of the initial corpus in two ways:
> - We measure **how many pairs of erroneous agreement appear in the corpus** (and find that this is much larger than the number that can fit in the context window),
> - We measure **how many pairs of erroneous agreement are needed to produce systematic failures** (and find that even for fewer pairs than we test in the original version, we still recover many failures).
>
> Overall, we hope these help alleviate the concern that our method is bottlenecked by the corpus, and include further details below.
>
> 1. _The number of erroneous agreements in each corpus._ While we only use 150 pairs of erroneous agreement in the prompt (due to the context window), we scrape 33922 pairs of erroneous agreements from SNLI (using 157351 examples to make pairs), and 2131440 pairs of erroneous agreement from MS-COCO (using 616767 examples to make pairs). Intuitively, even relatively small corpora may produce many examples of erroneous agreement, since the number of possible pairs scales quadratically with the size of the corpus. We think the main bottleneck is the *context window of the model*, which only allows us to input ~150 pairs of examples, rather than the corpus itself.
> 2. _The number of erroneous agreements the categorizer needs to generate failures._  In our paper, we show that MultiMon needs some pairs to produce high-quality systematic failures [184 - 187], and produces many systematic failures with 150 pairs (Figure 3). To augment this, we additionally try giving the language model the top-k pairs from the scraping step (using MS-COCO), then generate systematic failures from them. We report the results in Table 1 of the supplement, and find that the system still finds some systematic failures with even 10 examples and 70% of the systematic failures with just 80 examples. These results suggest that even if the corpus does not have many pairs of erroneous agreement (5 orders of magnitude fewer for MS-COCO), MultiMon still produces failures.
>
> ---
>
> *The authors also do not provide examples of datasets that can be used as initial corpuses for their system (except for COCO that they use for experimentation)*
>
> In this work we study using SNLI in addition to MS-COCO as our corpus datasets, and find that different corpora produce different failures [Figure 3. Main body, Section C.2 Appendix]. Beyond these two corpora, any reasonably large dataset containing sentence descriptions would be suitable, such as MNLI [1], Conceptual Captions [2], or Flickr30k [3]. In practice, we expect that model providers (e.g. Stable Difusion, MidJourney) would simply collect inputs that real users submit, then input those to MultiMon as the corpus.
>
> ---
>
> *Their system is easy to use for an adversary as well (dual use) … [as] given the heavy reliance on the initial corpus on the types of failures, the proposed system generates, it's hard to be convinced that MultiMon is useless to the adversary (they can always use a better/diverse initial corpus) even if the evaluator had used the system to initially to do robust testing.*
>
> Thanks for raising this point; while adversaries could try using a different corpus than the evaluator, we think evaluators (defenders) are more likely to have better corpora, as they can see what queries users submit. For example, StabilityAI and MidJourney will have access to a large volume of submitted queries that they can use as a corpus, while the adversary does not. However, in scenarios where the adversary does have a better corpus, there could be an imbalance; we will add this to our discussion of risks.
>
> ---
>
> [1] Williams, Adina, Nikita Nangia, and Samuel R. Bowman. "A broad-coverage challenge corpus for sentence understanding through inference." arXiv preprint arXiv:1704.05426 (2017).
>
> [2] Sharma, Piyush, et al. "Conceptual captions: A cleaned, hypernymed, image alt-text dataset for automatic image captioning." Proceedings of the 56th Annual Meeting of the Association for Computational Linguistics (Volume 1: Long Papers). 2018.
>
> [3] Plummer, Bryan A., et al. "Flickr30k entities: Collecting region-to-phrase correspondences for richer image-to-sentence models." Proceedings of the IEEE international conference on computer vision. 2015.

---

### Official Review · Reviewer_VAzu · 2023-07-06

**Soundness:** 3 good
**Presentation:** 4 excellent
**Contribution:** 3 good
**Rating:** 6
**Confidence:** 2

**Summary:**

The authors of the paper observe that CLIP is employed by many generative multi-modal models, as such, they seek to generate failing cases by determining textual inputs that are close within CLIP embedding space while being distant in DistillRoBERTa. They then determine the type of failing cases via an LLM primed to categorise the failing cases into the type of failure (negation, temporal, attribute differences etc.). Finally, they use the categories and the original examples and generalise to new additional failure cases via an additional LLM prompt. They show how different dataset-LLM pairs uncover different failure cases in the generative multimodal models. The later two phases of the proposed approach have been shown to depend on the LLM used (GPT-3.5 performing worse than GPT-4 or Claude). The authors also show the failures that stem from CLIP flow downstream, demonstrating the issue across text-to-{image, 3D, video} models.

**Strengths:**

- Clear framework and hypothesis (similarity in CLIP with dissimilarity in textual embeddings are likely to be problematic inputs).
- Exploring the specialisation of MultiMon for a specific task, here demonstrated through self-driving, showing how a user of the framework may focus on a subset/subspace of interest for input generation
- Exploring the impact of corpus-LLM combinations, showing how the approach may be improved later

**Weaknesses:**

- Some magical numbers used in the prompt are unclear (41, 82). Are these due to prompt limits, or some other limit?
- The dependence on a dataset for scraping and chaining LLMs to categorise and generalise may create cascade failures in MultiMon.
- Sensitivity to prompts can be an issue, however, the prompts being shared mitigate this issue to some degree.
- Part of the difficulty in prompt generation is abstracted away, discussing how the prompt could be augmented for larger context window LLMs could be useful in the Appendix, perhaps related to the first point here.

**Questions:**

The work seems to be going towards a direction that is similar to fuzz-testing. Did the authors consider how their approach compares to fuzz-testing (specialised to multimodal models)? Some of the steps are already present, input generation (based on finding candidates that are similar in one embedding and different in another) and generalisation/generation through LLMs. The missing step seems to be a fitness function, say visual similarity performed in an automatic manner if the target domain is diffusion models.

### Discussion Phase
The authors have clarified the issues related to the first four weaknesses above clearly, although there are some lingering concerns despite the empirical results, I do not feel they are critical.

As for the fuzzing point. I feel I was not sufficiently clear with the direction it is applied, however, it was also clearer during the discussion phase that this would be solidly outside of the scope of the paper and instead future work.

**Limitations:**

The work and failure cases can be used to bypass guard rails and filters employed by models that use CLIP. The authors do demonstrate this capability, however, this is done in the hope of enabling the detection of such issues before models are in production and hence this disclosure seems prudent. This is addressed, but only briefly in the additional material in the section demonstrating this capability.

---

> ### Author Rebuttal · Authors · 2023-08-09
>
> We thank the reviewer for the helpful comments and suggestions! We appreciate that you found our framework clear and liked our steering experiments.  We respond to your questions and comments below:
>
> ---
>
> *Some magical numbers used in the prompt are unclear (41, 82). Are these due to prompt limits, or some other limit?*
>
> Yes, we set the numbers in the prompt to 41 because we find empirically that GPT-4 can output at most 41 pairs of failure instances in one response; sorry we did not specify this initially! We will add it to subsequent versions of our work.
>
> ---
>
> *The dependence on a dataset for scraping and chaining LLMs to categorise and generalise may create cascade failures in MultiMon.*
>
> This is an interesting point; at least empirically, we find that MultiMon reliably produces failures across all combinations of two corpus datasets (SNLI and MS-COCO) and three LLM (GPT-3.5, GPT-4 and Claude v1.3) [Section 4.1, Appendix C.2, C.3, and C.4]. One hypothesis for why we don’t see cascading failures is we only have three steps: steering, categorizing, and generating, so even if all steps fail slightly, the aggregate of these three failures may not be significant.
>
> ---
>
> *Sensitivity to prompts could be an issue + generating prompts may be difficult.*
>
> Though prompt sensitivity is an issue for other tasks, we find that our prompt produces good performance across all six of the model-corpora combinations that we test, and we expect the same prompt will continue to work for longer context windows. Nevertheless, we think there could even be room to improve MultiMon with more careful prompt selection, and will add this to the discussion.
>
> ---
>
> *Did the authors consider how their approach compares to fuzz-testing on multimodal models?*
>
>
> Thanks for your question; while our work at a high level is similar to fuzz testing (i.e., we scrape for candidates and adapt them), the inputs we choose are “in-distribution” (since they come from a corpus), and MultiMon is one shot: we simply scrape, categorize, and generate without iterating. Studying what insights from fuzz testing port well to identifying failures in ML systems seems like an interesting direction for subsequent work.

---

> > ### Comment · Reviewer_VAzu · 2023-08-10
> >
> > Thank you for the clarifications!
> >
> > At a glance, the low stacked error as a reason for no cascading failures makes sense.
> >
> > As for the fuzzing part, iteration was not included from the start to direct fuzzing towards interesting inputs. I was comparing more on the multiple samples from some assumed distribution. I agree that this is more for subsequent work, but there is perhaps potential where MultiMon acts as a sort of "prior" or starting distribution before iteration. In particular, the steering mechanism seems quite apt for this direction.

---

### Official Review · Reviewer_vAM5 · 2023-07-07

**Soundness:** 3 good
**Presentation:** 4 excellent
**Contribution:** 4 excellent
**Rating:** 7
**Confidence:** 4

**Summary:**

This paper proposes a system which can find and identify the systematic failures of existing multi-modal models. It measures the failures by comparing whether the model produces unexpected similar outputs for different inputs. It finds a lot of failures of the CLIP text encoder and utilizes the language model to categorizes them. this paper also discusses the effect of the found failures on different downstream tasks including text-to-image generation. Those failures can be helpful for repairing the multi-model models.

**Strengths:**

1. This paper is well-written,  and the presentation is clear and easy to understand.
2. The proposed method to detect failures in multi-modal models is simple and effective. It can automatically find and categorize various failures in multi-model systems.
3. The found failures are useful for analyzing the limitation of the multi-modal models and downstream models such as text-to-image generation models. It would provide a clear direction to refine those models.

**Weaknesses:**

1. It would be helpful if the paper conducts further analysis of the reason of those failures, which may point out some systematic problems of the design of existing multi-modal models.
2. This paper only detect and categorize the failures of the CLIP text encoder. This paper can add more experiments to evaluate the proposed methods on different multi-modal models and tasks.

**Questions:**

Can the proposed methods directly detect the failures of the multi-modal models when dealing image inputs? For example, detecting that the CLIP image encoder provides similar embeddings for different images.

**Limitations:**

The authors have addressed the technical limitation of this paper. There is no obvious negative societal impact of this work.

---

> ### Author Rebuttal · Authors · 2023-08-09
>
> Thank you for your helpful comments and interesting questions! We appreciate that you found our work well-written, clear, and useful for analyzing the limitation of multimodal models. We respond to your questions below.
>
> ---
>
> *This paper only detects and categorizes the failures of the CLIP text encoder. This paper can add more experiments to evaluate the proposed methods on different multi-modal models and tasks*
>
> Thank you for your suggestion! Based of it, we test how well our system can find failures on multimodal models that do not use CLIP in two settings:
>
> 1. **Transfer**. We show that the inputs we find using CLIP often directly produce failures in T5-based text-to-image models; 70.8% of the inputs used to generate figures in the main body or the appendix produce failures on DeepFloyd, a text-to-image model that uses T5 to embed inputs, rather than CLIP. [Figure 1, supplementary material]
>
> 2. **Generation from scratch**. We applied MultiMon on the T5 model from scratch (i.e., replaced CLIP with T5 throughout the whole pipeline) and found some systematic failures of the T5-based model that the models we study do not have [Figure 2, supplementary material].
>
> We go into further on the generation experiment below.
>
> **Generation details.** To find systematic failures of DeepFloyd, from scratch, we repeat the entire MultiMon pipeline from Section 3, using T5 embeddings instead of CLIP (using our existing code, this took under three hours.) We find many overlapping systematic failures but some new ones. For example, some new systematic failures that MultiMon outputs are:
>
> - *The model fails to comprehend the use of pronouns properly. The sentences are similar, but the change of subject affects the visual representation significantly*
> - *The model fails to distinguish the time of day. This is critical in visual representation, as these times would significantly change the lighting, color scheme, and potentially the activity depicted in the image.*
>
> Overall, the systematic failures we find with MultiMon on T5 have an average success rate of 77.3%. We have also attached some inputs that MultiMon generates using T5 along with their images generated with DeepFloyd [Figure 2, supplementary material].
>
> ---
>
> *It would be helpful if the paper can conduct further analysis of the reason for these failures, which may point out some systematic problems of the design of existing multimodal models?*
>
> Thanks for your suggestion! Based on the construction of MultiMon, we can say that the failures we uncover are caused entirely by the text-encoder; this means that no matter how effectively someone trains a diffusion model on top of these embeddings, failures will remain (so the embedding model itself needs to be fixed or changed) [298 - 305]. Beyond that, it is hard to come up with causal reasons for the failures; one hypothesis is that failing to encode negation, numerical differences and bag-of-words arise because there are insufficient discriminatory examples during pretraining, while another is that the dimensionality of the embeddings is too small (768) to encode all relevant features that appear in text. Robustly testing hypotheses like these is an important direction for subsequent work.
>
> ---
>
> *Can the proposed methods directly detect the failures of the multi-modal models when dealing with image inputs? For example, detecting that the CLIP image encoder provides similar embeddings for different images.*
>
>
> Conceptually, we think the MultiMon framework can be used to find failures of the CLIP image encoder: for scraping, we could take a dataset of images, and find examples of erroneous agreement by comparing against a pretrained image encoder. The challenge comes at the “categorization” step; since we use language models to identify systematic failures, the inputs must be text, not images. However, as text-guided image-to-text systems improve, an multimodal text + image-to-text model (like Google’s Bard [1], Llava [2], or multimodal GPT-4 [3]) could substitute as a categorizer, and a text-to-image model could serve as the generator. We think this highlights the generality of our framework, and would be an interesting direction for subsequent work.
>
> ---
>
> [1] Google, An important next step on our AI journey, 2023
>
> [2] Liu, Haotian, et al. "Visual instruction tuning." arXiv preprint arXiv:2304.08485 (2023).
>
> [3] OpenAI. Gpt-4 technical report, 2023.

---

### Author Rebuttal · Authors · 2023-08-09


We thank all of the reviewers for their feedback on our work. Reviewers liked that our framework is clear (VAzu, ATN5), simple and effective (vAM5), flexible with plug-and-play components (dXH6), and steerable towards certain subdomains (VAzu, dXH6), and could have staying power, saying “approaches like this one will only become more important” (ATN5).

Multiple Reviewers (vAM5, ATN5) were interested in whether MultiMon can find failures in text-to-image models that are known to not use CLIP. In response, we test MultiMon on such systems in two settings.

1. **Transfer**. We show that the inputs we find using CLIP often directly produce failures in T5-based [1] text-to-image models; 70.8% of the inputs used to generate figures in the main body or the appendix produce failures on DeepFloyd [2], a diffusion model that uses T5 to embed inputs rather than CLIP. [Figure 1, supplementary material]

2. **Generation from scratch**. We applied MultiMon on the T5-based DeepFloyad from scratch (i.e., replaced CLIP with T5 throughout the whole pipeline) and find many systematic failures, some of which the CLIP model does not have (e.g., T5 struggles to encode pronouns properly) [Figure 2, supplementary material].

We respond to individual reviewer comments below.

[1] Raffel, Colin, et al. "Exploring the limits of transfer learning with a unified text-to-text transformer." The Journal of Machine Learning Research 21.1 (2020): 5485-5551.

[2] Alex, Misha, et al. DeepFloyd IF by DeepFloyd Lab at StabilityAI. https://github.com/deep-floyd/IF, (2023)

---

### Decision · Program_Chairs · 2023-09-21

**Decision:**

Accept (poster)

**Comment:**

I will recommend this paper for acceptance, because reviewers felt the paper is well-written (vAM5, ATN5), the framework and hypothesis are clear (VAzu), the method simple and effective (vAM5, dXH6), and the proposed system is flexible (dXH6).

Engagement in rebuttal was strong and positive. After the discussion period there few concerns remaining.

Reviewers did uncover some opportunities for improvement: it would be nice to analyze the reason for the failures (vAM5), they worried the approach is narrowly focused on CLIP (vAM5) but rebuttal added additional context beyond CLIP. Reviewers noted that some prompt decisions were unclear (VAzu), there were possibilities of cascading failures (VAzu), the work had heavy reliance on anexisting corpus (dXH6), and it would’ve been nice to have external annotators (ATN5). However, reviewers felt this work meets the bar for acceptance.

Recommendations for the next version of this work: that prompt decisions be clarified (see VAzu), difficulty in generating prompts should be discussed (VAzu), discussion of and comparison to fuzz-testing in the paper is recommended (VAzu stated “I agree that this is more for subsequent work"), possible dual use should be mentioned (VAzu, dHX6) and examples of good starting datasets could be included (dXH6). ATN5 felt it could be interesting to consider automatically fixing failures and they also suggested that the authors consider pulling figures from appendix to enhance readability.